# Multiscale Dictionary Learning for Estimating Conditional Distributions

**Francesca Petralia**
Department of Genetics and Genomic Sciences
Icahn School of Medicine at Mt Sinai
New York, NY 10128, U.S.A.
`francesca.petralia@mssm.edu`

**Joshua Vogelstein**
Child Mind Institute
Department of Statistical Science
Duke University
Durham, North Carolina 27708, U.S.A.
`jo.vo@duke.edu`

**David B. Dunson**
Department of Statistical Science
Duke University
Durham, North Carolina 27708, U.S.A.
`dunson@stat.duke.edu`

## Abstract

Nonparametric estimation of the conditional distribution of a response given high-dimensional features is a challenging problem. It is important to allow not only the mean but also the variance and shape of the response density to change flexibly with features, which are massive-dimensional. We propose a multiscale dictionary learning model, which expresses the conditional response density as a convex combination of dictionary densities, with the densities used and their weights dependent on the path through a tree decomposition of the feature space. A fast graph partitioning algorithm is applied to obtain the tree decomposition, with Bayesian methods then used to adaptively prune and average over different sub-trees in a soft probabilistic manner. The algorithm scales efficiently to approximately one million features. State of the art predictive performance is demonstrated for toy examples and two neuroscience applications including up to a million features.

## 1    Introduction

Massive datasets are becoming an ubiquitous by-product of modern scientific and industrial applications. These data present statistical and computational challenges because many previously developed analysis approaches do not scale-up sufficiently. Challenges arise because of the ultra high-dimensionality and relatively low sample size. Parsimonious models for such big data assume that the density in the ambient space concentrates around a lower-dimensional (possibly nonlinear) subspace. A plethora of methods are emerging to estimate such lower-dimensional subspaces [1, 2].

We are interested in using such lower-dimensional embeddings to obtain estimates of the conditional distribution of some target variable(s). This *conditional density estimation* setting arises in a number of important application areas, including neuroscience, genetics, and video processing. For example, one might desire automated estimation of a predictive density for a neurologic  phenotype of interest, such as intelligence, on the basis of available data for a patient including neuroimaging. The challenge is to estimate the probability density function of the phenotype *nonparametrically* based on a $10^6$ dimensional image of the subject's brain. It is crucial to avoid parametric assumptions on the density, such as Gaussianity, while allowing the density to change flexibly with predictors. Otherwise, one can obtain misleading predictions and poorly characterize predictive uncertainty.

There is a rich machine learning and statistical literature on conditional density estimation of a response $y \in \mathcal{Y}$ given a set of features (predictors) $x = (x_1, x_2, \ldots, x_p)^\mathsf{T} \in \mathcal{X} \subseteq \mathbb{R}^p$. Common approaches include hierarchical mixtures of experts [3, 4], kernel methods [5, 6, 7], Bayesian finite mixture models [8, 9, 10] and Bayesian nonparametrics [11, 12, 13, 14]. However, there has been limited consideration of scaling to large $p$ settings, with the variational Bayes approach of [9] being a notable exception. For dimensionality reduction, [9] follow a greedy variable selection algorithm. Their approach does not scale to the sized applications we are interested in. For example, in a problem with $p = 1,000$ and $n = 500$, they reported a CPU time of 51.7 minutes for a single analysis. We are interested in problems with $p$ having many more orders of magnitude, requiring a faster computing time while also accommodating flexible nonlinear dimensionality reduction (variable selection is a limited sort of dimension reduction). To our knowledge, there are no nonparametric density regression competitors to our approach, which maintain a characterization of uncertainty in estimating the conditional densities; rather, all sufficiently scalable algorithms provide point predictions and/or rely on restrictive assumptions such as linearity.

In big data problems, scaling is often accomplished using divide-and-conquer techniques. However, as the number of features increases, the problem of finding the best splitting attribute becomes intractable, so that CART, MARS and multiple tree models cannot be efficiently applied. Similarly, mixture of experts becomes computationally demanding, since both mixture weights and dictionary densities are predictor dependent. To improve efficiency, sparse extensions relying on different variable selection algorithms have been proposed [15]. However, performing variable selection in high dimensions is effectively intractable: algorithms need to efficiently search for the best subsets of predictors to include in weight and mean functions within a mixture model, an NP-hard problem [16].

In order to efficiently deal with massive datasets, we propose a novel multiscale approach which starts by learning a multiscale dictionary of densities. This tree is efficiently learned in a first stage using a fast and scalable graph partitioning algorithm applied to the high-dimensional observations [17]. Expressing the conditional densities $f(y|x)$ for each $x \in \mathcal{X}$ as a convex combination of coarse-to-fine scale dictionary densities, the learning problem in the second stage estimates the corresponding multiscale probability tree. This is accomplished in a Bayesian manner using a novel multiscale stick-breaking process, which allows the data to inform about the optimal bias-variance tradeoff; weighting coarse scale dictionary densities more highly decreases variance while adding to bias. This results in a model that borrows information across different resolution levels and reaches a good compromise in terms of the bias-variance tradeoff. We show that the algorithm scales efficiently to millions of features.

## 2   Setting

Let $X : \Omega \to \mathcal{X} \subseteq \mathbb{R}^p$ be a $p$-dimensional Euclidean vector-valued predictor random variable, taking values $x \in \mathcal{X}$, with a marginal probability distribution $f_X$. Similarly, let $Y : \Omega \to \mathcal{Y}$ be a target-valued random variable (e.g., $\mathcal{Y} \subseteq \mathbb{R}$). For inferential expedience, we posit the existence of a latent variable $\eta : \Omega \to \mathcal{M} \subseteq \mathcal{X}$, where $\mathcal{M}$ is only $d$ "dimensional" and $d \ll p$. Note that $\mathcal{M}$ need not be a linear subspace of $\mathcal{X}$, rather, $\mathcal{M}$ could be, for example, a union or affine subspaces, or a smooth compact Riemannian manifold. Regardless of the nature of $\mathcal{M}$, we assume that we can approximately decompose the joint distribution as follows, $f_{X,Y,\eta} = f_{X,Y|\eta} f_\eta = f_{Y|X,\eta} f_{X|\eta} f_\eta \approx f_{Y|\eta} f_{X|\eta} f_\eta$. Hence, we assume that the *signal* approximately concentrates around a low-dimensional latent space, $f_{Y|X,\eta} = f_{Y|\eta}$. This is a much less restrictive assumption than the commonplace assumption in manifold learning that the marginal distribution $f_X$ concentrates around a low-dimensional latent space.

To provide some intuition for our model, we provide the following concrete example where the distribution of $y \in \mathbb{R}$ is a Gaussian function of the coordinate $\eta \in \mathcal{M}$ along the swissroll, which is embedded in a high-dimensional ambient space. Specifically, we sample the manifold coordinate, $\eta \sim U(0, 1)$. We sample $x = (x_1, \ldots, x_p)^\mathsf{T}$ as follows

$$x_1 = \eta \sin(\eta) \quad ; \quad x_2 = \eta \cos(\eta) \quad ; \quad x_r \sim \mathcal{N}(0, 1) \quad r \in \{3, \ldots, p\}$$

Finally, we sample $y$ from $\mathcal{N}(\mu(\eta), \sigma(\eta))$. Clearly, $x$ and $y$ are conditionally independent given $\eta$, which is the low-dimensional signal manifold. In particular, $x$ lives on a swissroll embedded in a

$p$-dimensional ambient space, but $y$ is only a function of the coordinate $\eta$ along the swissroll $\mathcal{M}$. The left panels of Figure 1 depict this example when $\mu(\eta) = \eta$ and $\sigma(\eta) = \eta + 1$.

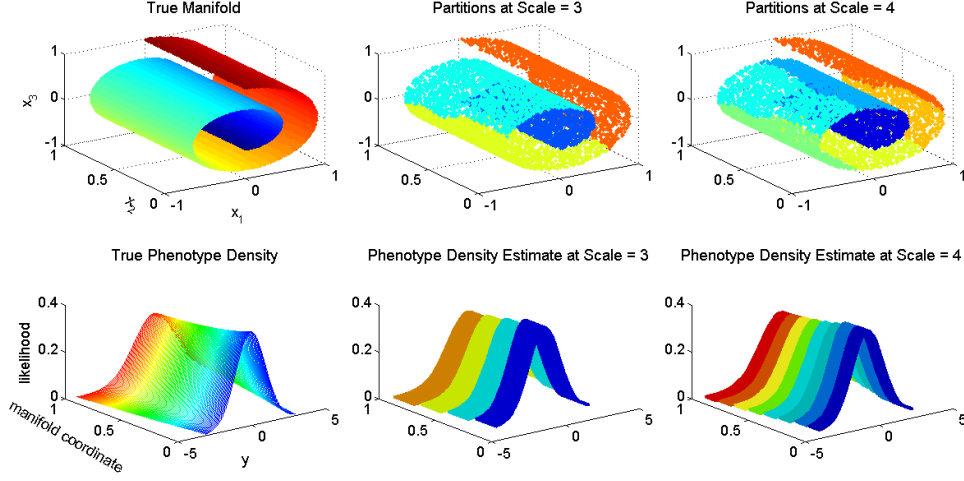

Figure 1: Illustration of our generative model and algorithm on a swissroll. The top left panel shows the manifold $\mathcal{M}$ (a swissroll) embedded in a $p$-dimensional ambient space, where the color indicates the coordinate along the manifold, $\eta$ (only the first 3 dimensions are shown for visualization purposes). The bottom left panel shows the distribution of $y$ as a function of $\eta$, in particular, $f_{Y|\eta} = \mathcal{N}(\eta, \eta + 1)$. The middle and right panels show our estimates of $f_{Y|\eta}$ at scales 3 and 4, respectively, which follow from partitioning our data. Sample size was $n = 10,000$.

## 3 Goal

Our goal is to develop an approach to learn about $f_{Y|X}$ from $n$ pairs of observations that we assume are exchangeable samples from the joint distribution, $(x_i, y_i) \sim f_{X,Y} \in \mathcal{F}$. Let $\mathcal{D}^n = \{(x_i, y_i)\}_{i \in [n]}$, where $[n] = \{1, \dots, n\}$. More specifically, we seek to obtain a posterior over $f_{Y|X}$. We insist that our approach satisfies several desiderata, including most importantly: (i) scales up to $p \approx 10^6$ in reasonable time, (ii) yields good empirical results, and (iii) automatically adapts to the complexity of the data corpus. To our knowledge, no extant approach for estimating conditional densities or posteriors thereof satisfies even our first criterion.

## 4 Methodology

### 4.1 Ms. Deeds Framework

We propose here a general modular approach which we refer to as multiscale dictionary learning for estimating conditional distributions ("Ms. Deeds"). Ms. Deeds consists of two components: (i) a tree decomposition of the space, and (ii) an assumed form of the conditional probability model.

**Tree Decomposition** A tree decomposition $\tau$ yields a multiscale partition of the data or the ambient space in which the data live. Let $(\mathcal{W}, \rho_W, F_W)$ be a measurable metric space, where $F_W$ is a Borel probability measure, $\mathcal{W}$, and $\rho_W: \mathcal{W} \times \mathcal{W} \to \mathbb{R}$ is a metric on $\mathcal{W}$. Let $B_r^{\mathcal{W}}(w)$ be the $\rho_W$-ball inside $\mathcal{W}$ of radius $r > 0$ centered at $w \in \mathcal{W}$. For example, $\mathcal{W}$ could be the data corpus $\mathcal{D}_n$, or it could be $\mathcal{X} \times \mathcal{Y}$. We define a tree decomposition as in [2, 18]. A partition tree $\tau$ of $\mathcal{W}$ consists of a collection of cells, $\tau = \{C_{j,k}\}_{j \in \mathbb{Z}, k \in \mathcal{K}_j}$. At each scale $j$, the set of cells $C_j = \{C_{j,k}\}_{k \in \mathcal{K}_j}$ provides a disjoint partition of $\mathcal{W}$ almost everywhere. We define $j = 0$ as the root node. For each $j > 0$, each set has a unique parent node. Denote

$$A_{j,k} = \{(j', k') : C_{j,k} \subseteq C_{j',k'}, \ j' < j\} \ , \quad D_{j,k} = \{(j', k') : C_{j',k'} \subseteq C_{j,k}, \ j' > j\}$$

respectively the ancestors and the descendants of node $(j, k)$.

Unlike classical harmonic theory which presupposes $\tau$ (e.g., in wavelets [19]), we choose to learn $\tau$ from the data. Previously, Chen et al. [18] developed a multiscale measure estimation strategy, and proved that there exists a scale $j$ such that the approximate measure is within some bound of the true measure, under certain relatively general assumptions. We decided to simply partition the $x$'s, ignoring the $y$'s in the partitioning strategy. Our justification for this choice is as follows. First, sometimes there are many different $y$'s for many different applications. In such cases, we do not want to bias the partitioning to any specific $y$'s, all the more so when new unknown $y$'s may later emerge. Second, because the $x$'s are so much higher dimensional than the $y$'s in our applications of interest, the partitions would be dominated by the $x$'s, unless we chose a partitioning strategy that emphasized the $y$'s. Thus, our strategy mitigates this difficulty (while certainly introducing others).

Given that we are going to partition using only the $x$'s, we still face the choice of precisely how to partition. A fully Bayesian approach would construct a large number of partitions, and integrate over them to obtain posteriors. However, such a fully Bayesian strategy remains computationally intractable at scale, so we adopt a hybrid strategy. Specifically, we employ METIS [17], a well-known relatively efficient multiscale partitioning algorithm with demonstrably good empirical performance on a wide range of graphs. Given $n$ observations, i.e. $x_i = (x_{i1}, \ldots, x_{ip})^{\mathsf{T}} \in \mathcal{X}$ for $i \in [n]$, the graph construction follows via computing all pairwise distances using $\rho(x_u, x_v) = \|\tilde{x}_u - \tilde{x}_v\|_2$, where $\tilde{x}$ is the whitened $x$ (i.e., mean subtracted and variance normalized). We let there be an edge between $x_u$ and $x_v$ whenever $e^{-\rho(x_u,x_v)^2} > t$, where $t$ is some threshold chosen to elicit the desired sparsity level. Applying METIS recursively on the graph constructed in this way yields a single tree (see supplementary material for further details).

**Conditional Probability Model**  Given the tree decomposition of the data, we place a nonparametric prior over the tree. Specifically, we define $f_{Y|X}$ as

$$f_{Y|X} = \sum_{j \in \mathbb{Z}} \pi_{j,k_j(x)} f_{j,k_j(x)}(y|x) \tag{1}$$

where $k_j(x)$ is the set at scale $j$ where $x$ has been allocated and $\pi_{j,k_j}(x)$ are weights *across scales* such that $\sum_{j \in \mathbb{Z}} \pi_{j,k_j(x)} = 1$. We let weights in Eq. (1) be generated by a stick-breaking process [20]. For each node $C_{j,k}$ in the partition tree, we define a stick length $V_{j,k} \sim \text{Beta}(1, \alpha)$. The parameter $\alpha$ encodes the complexity of the model, with $\alpha = 0$ corresponding to the case in which $f(y|x) = f(y)$. The stick-breaking process is defined as

$$\pi_{j,k} = V_{j,k} \prod_{(j',k') \in A_{j,k}} [1 - V_{j',k'}], \tag{2}$$

where $\sum_{(j',k') \in A_{j,k}} \pi_{j',k'} = 1$. The implication of this is that each scale within a path is weighted to optimize the bias/variance trade-off across scales. We refer to this prior as a *multiscale stick-breaking process*. Note that this Bayesian nonparametric prior assigns a positive probability to all possible paths, including those not observed in the training data. Thus, by adopting this Bayesian formulation, we are able to obtain posterior estimates for any newly observed data, regardless of the amount and variability of training data. This is a pragmatically useful feature of the Bayesian formulation, in addition to the alleviation of the need to choose a scale [18].

Each $f_{j,k}$ in Eq. (1) is an element of a family of distributions. This family might be quite general, e.g., all possible conditional densities, or quite simple, e.g., Gaussian distributions. Moreover, the family can adapt with $j$ or $k$, being more complex at the coarser scales (for which $n_{j,k}$'s are larger), and simpler for the finer scales (or partitions with fewer samples). We let the family of conditional densities for $y$ be Gaussian for simplicity, that is, we assume that $f_{j,k} = \mathcal{N}(\mu_{j,k}, \sigma_{j,k})$ with $\mu_{j,k} \in \mathbb{R}$ and $\sigma_{j,k} \in \mathbb{R}^+$. Because we are interested in posteriors over the conditional distribution $f_{Y|X}$, we place relatively uninformative but conjugate priors on $\mu_{j,k}$ and $\sigma_{j,k}$, specifically, assuming the $y$'s have been whitened and are unidimensional, $\mu_{j,k} \sim \mathcal{N}(0, 1)$ and $\sigma_{j,k} = \mathcal{IG}(a, b)$. Obviously, other choices, such as finite or infinite mixtures of Gaussians are also possible for continuous valued data.

### 4.2  Inference

We introduce the latent variable $\ell_i \in \mathbb{Z}$, for $i = [n]$, denoting the multiscale level used by the $i^{th}$ observation. Let $n_{j,k}$ be the number of observations in $C_{j,k}$. Let $k_h(x_i)$ be a variable indicating the

set at level $h$ where $x_i$ has been allocated. Each Gibbs sampler iteration can be summarized in the following steps:

(i) Update $\ell_i$ by sampling from the multinomial full conditional:

$$\Pr(\ell_i = j \,|\, \cdot) = \pi_{j,k_j(x_i)} f_{j,k_j(x_i)}(y_i|x_i) / \sum_{s \in \mathbb{Z}} \pi_{s,k_s(x_i)} f_{s,k_s(x_i)}(y_i|x_i)$$

(ii) Update stick-breaking random variable $V_{j,k}$, for any $j \in \mathbb{Z}$ and $k \in \mathcal{K}_j$, from $\mathrm{Beta}(\beta', \alpha')$ with $\beta' = 1 + n_{j,k}$ and $\alpha' = \alpha + \sum_{(r,s) \in D_{j,k}} n_{r,s}$.

(iii) Update $\mu_{j,k}$ and $\sigma_{j,k}$, for any $j \in \mathbb{Z}$ and $k \in \mathcal{K}_j$, by sampling from

$$\mu_{j,k} \sim \mathcal{N}\left(\upsilon_{j,k}\nu_{j,k}\bar{y}_{j,k}, \upsilon_{j,k}\right), \quad \sigma_{j,k} \sim \mathcal{IG}\left(a_\sigma, b + 0.5\sum_{i \in \mathcal{I}_{j,k}}\left(y_i - \mu_{j,k}\right)^2\right)$$

where $\upsilon_{j,k} = (1 + \nu_{j,k})^{-1}$, $\nu_{j,k} = n_{j,k}/\sigma_{j,k}$ $a_\sigma = a + n_{j,k}/2$, $\bar{y}_{j,k}$ being the average of the observations $\{y_i\}$ allocated to cell $C_{j,k}$ and $\mathcal{I}_{j,k} = \{i : \ell_i = j, x_i \in C_{j,k}\}$.

To make predictions, the Gibbs sampler was run with up to $20,000$ iterations, including a burn-in of $1,000$ (see Supplementary material for details). Gibbs sampler chains were stopped testing normality of normalized averages of functions of the Markov chain [21]. Parameters $(a, b)$ and $\alpha$ involved in the prior density of parameters $\sigma_{j,k}$'s and $V_{j,k}$'s were set to $(3, 1)$ and $1$, respectively. All predictions used a leave-one-out strategy.

### 4.3 Simulation Studies

In order to assess the predictive performance of the proposed model, we considered the four different simulation scenarios described below:

**(1) Nonlinear Mixture** We first consider a relatively simple yet nonlinear joint model, with a conditional Gaussian distribution $y|\eta \sim |\eta|\mathcal{N}(\mu_1, \sigma_1) + (1 - |\eta|)\mathcal{N}(\mu_2, \sigma_2)$, a marginal distribution for each dimension of $x$, $x_r|\eta \sim \mathcal{N}(\eta, \sigma_x)$, $r \in \{1, 2, \ldots, p\}$, and a uniform distribution over the latent manifold $\eta \sim \sin(U(0, c))$. In the simulations we let $(\mu_1, \sigma_1) = (-2, 1)$, $(\mu_2, \sigma_2) = (2, 1)$, $\sigma_x = 0.1$, and $c = 20$, and $p = 1000$. Thus, $f_{Y|X}$ is a highly nonlinear function of $x$, and even $\eta$, and $x$ is high-dimensional.

**(2) Swissroll** We then return to the swissroll example of Figure 1; in Figure 3 we show results for $(\mu, \sigma) = (\eta, 1)$.

**(3) Linear Subspace** Letting $\Gamma \in \mathbb{R}^{p+1 \times q}$ and $\Theta$ be a $q \times d$ "diagonal" matrix (meaning all entires other than the first $d < q$ elements of the diagonal are zero), we assume the following model: $Y, X|\eta \sim \mathcal{N}_{p+1}(\Gamma\Theta\eta, I)$, where $\Gamma \sim \mathcal{S}_{p+1,d}$ indicates $\Gamma$ is uniformly sampled from the set of all orthonormal $d$ frames in $\mathbb{R}^{p+1}$ (a Stiefel manifold), $\theta_{ii} \sim \mathcal{IG}(a_\theta, b_\theta)$ for $i \in \{1, \ldots, d\}$ and all other elements of $\Theta$ are zero, and $\eta \sim \mathcal{N}_d(0, I)$. In the simulation, we let $q = d = 5$, $(\alpha_\theta, \beta_\theta) = (1, 0.25)$.

**(4) Union of Linear Subspaces** This model is a direct extension of the linear subspace model, as it is a union of subspaces. We let the dimensionality of each subspace vary to demonstrate the generality of our procedure. Specifically, we assume $Y, X|\eta \sim \sum_{g=1}^G \omega_g \mathcal{N}_{p+1}(\Gamma_g\Theta_g\eta, I)$, $\omega \sim Dirichlet(\boldsymbol{\alpha})$, $\eta \sim \mathcal{N}_d(0, I)$, where $\Gamma \sim \mathcal{S}_{p+1,g}$ and $\Theta_g$ is "diagonal" with $\theta_{ii} \sim \mathcal{IG}(a_g, b_g)$ for $i \in \{1, \ldots, g\}$, and the remaining elements of $\Theta$ are zero. In the simulation, we let $G = 5$, $\boldsymbol{\alpha} = (1, \ldots, 1)^\mathsf{T}$, $(\alpha_g, \beta_g) = (\alpha_\theta, \beta_\theta)$ as above.

### 4.4 Neuroscience Applications

We assessed the predictive performance of the proposed method on two very different neuroimaging datasets. For all analyses, each variable was normalized by subtracting its mean and dividing by its standard deviation. The prior specification and Gibbs sampler described in §4.1 and 4.2 were utilized.

In the first experiment we investigated the extent to which we could predict creativity (as measured via the Composite Creativity Index [22]) via a structural connectome dataset collected at the Mind Research Network (data were collected as described in Jung et al. [23]). For each subject, we estimate a 70 vertex undirected weighted brain-graph using the Magnetic Resonance Connectome Automated Pipeline (MRCAP) [24] from diffusion tensor imaging data [25]. Because our graphs are

undirected and lack self-loops, we have a total of $p = \binom{70}{2} = 2,415$ potential weighted edges. The $p$-dimensional feature vector is defined by the natural logarithm of the vectorized matrix described above.

The second dataset comes from a resting-state functional magnetic resonance experiment as part of the Autism Brain Imaging Data Exchange [26]. We selected the Yale Child Study Center for analysis. Each brain-image was processed using the Configurable Pipeline for Analysis of Connectomes (CPAC) [27]. For each subject, we computed a measure of normalized power at each voxel called fALFF [28]. To ensure the existence of nonlinear signal relating these predictors, we let $y_i$ correspond to an estimate of overall head motion in the scanner, called mean framewise displacement (FD) computed as described in Power et al. [29]. In total, there were $p = 902,629$ voxels.

### 4.5 Evaluation Criteria

To compare algorithmic performance we considered $r_m^{\mathcal{A}}$ defined as $r_m^{\mathcal{A}} = \phi(MSB)/\phi(\mathcal{A})$, where $\phi$ is the quantity of interest (for example, CPU time in seconds or mean squared error), MSB is our approach and $\mathcal{A}$ is the competitor algorithm. To obtain mean-squared error estimates from MSB, we select our posterior mean as a point-estimate (the comparison algorithms do not generate posterior predictions, only point estimates). For each simulation scenario, we sampled multiple datasets and compute the *matched* distribution of $r_m^{\mathcal{A}}$. In other words, rather than running simulations and reporting the distribution of performance for each algorithm, we compare the algorithms per simulation. This provides a much more informative indication of algorithmic performance, in that we indicate the fraction of simulations one algorithm outperforms another on some metric. For each example, we sampled 20 datasets to obtain estimates of the distribution over $r_m^{\mathcal{A}}$. All experiments were performed on a typical workstation, Intel Core i7-2600K Quad-Core Processor with 8192 MB of RAM.

## 5 Results

### 5.1 Illustrative Example

The middle and right panels of Figure 1 depict the quality of partitioning and density estimation for the swissroll example described in §2, with the ambient dimension $p = 1000$ and the predictive manifold dimension $d = 1$. We sampled $n = 10^4$ samples for this illustration. At scale 3 we have 4 partitions, and at scale 4 we have 8 (note that the partition tree, in general, need not be binary). The top panels are color coded to indicate which $x_i$'s fall into which partition. Although imperfect, it should be clear that the data are partitioned very well. The bottom panels show the resulting estimate of the posteriors at the two scales. These posteriors are *piecewise constant*, as they are invariant to the manifold coordinate within a given partition.

To obviate the need to choose a scale to use to make a prediction, we choose to adopt a Bayesian approach and integrate across scales. Figure 2 shows the estimated density of two observations of model (1) with parameters $(\mu_1, \sigma_1) = (-2, 1)$, $(\mu_2, \sigma_2) = (2, 1)$, $\sigma_x = 0.1$, and $c = 20$ for different sample sizes. Posteriors of the conditional density $f_{Y|X}$ were computed for various sample sizes. Figure 2 suggests that our estimate of $f_{Y|X}$ approaches the true density as the number of observations in the training set increases. We are unable to compare our strategy for posterior estimation to previous literature because we are unaware of previous Bayesian approaches for this problem that scale up to problems of this size. Therefore, we numerically compare the performance of our point-estimates (which we define as the posterior mean of $\hat{f}_{Y|X}$) with the predictions of the competitor algorithms.

### 5.2 Quantitative Comparisons for Simulated Data

Figure 3 compares the numerical performance of our algorithm (MSB) with Lasso (black), CART (red), and PC regression (green) in terms of both mean-squared error (top) and CPU time (bottom) for models (2), (3), and (4) in the left, middle, and right panels respectively. These figures show relative performance on a *per simulation basis*, thus enabling a much more powerful comparison than averaging performance for each algorithm over a set of simulations. Note that these three simulations span a wide range of models, including nonlinear smooth manifolds such as the swissroll

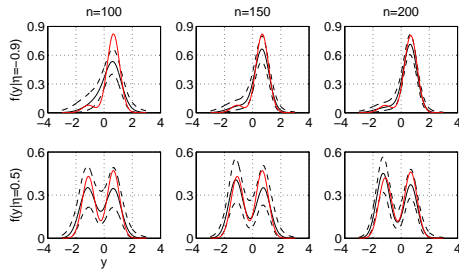

Figure 2: Illustrative example of model (1) suggesting that our posterior estimates of the conditional density are converging as $n$ increases even when $f_{Y|\eta}$ is highly nonlinear and $f_{X|\eta}$ is very high-dimensional. True (red) and estimated (black) density (50th percentile: solid line, 2.5th and 97.5th percentiles: dashed lines) for two data positions along the manifold (top panels: $\eta \approx -0.9$, bottom panels: $\eta \approx 0.5$) considering different training set sizes.

(model 2), relatively simple linear subspace manifolds (model 3), and a union of linear subspaces model (model 4; which is neither linear nor a manifold).

In terms of predictive accuracy, the top panels show that for all three simulations, in every dimensionality that we considered—including $p = 0.5 \times 10^6$—MSB is more accurate than either Lasso, CART, or PC regression. Note that this is the case even though MSB provides much more information about the posterior $f_{Y|X}$, yielding an entire posterior over $f_{Y|X}$, rather than merely a point estimate.

In terms of computational time, MSB is much faster than the competitors for large $p$ and $n$, as shown in the bottom three panels. The supplementary materials show that computational time for MSB is relatively constant as a function of $p$, whereas Lasso's computational time grows considerably with $p$. Thus, for large enough $p$, MSB is significantly faster that Lasso. MSB is faster than CART and PC regression for all $p$ and $n$ under consideration. Thus, it is clear from these simulations that MSB has better scaling properties—in terms of both predictive accuracy and computational time—than the competitor methods.

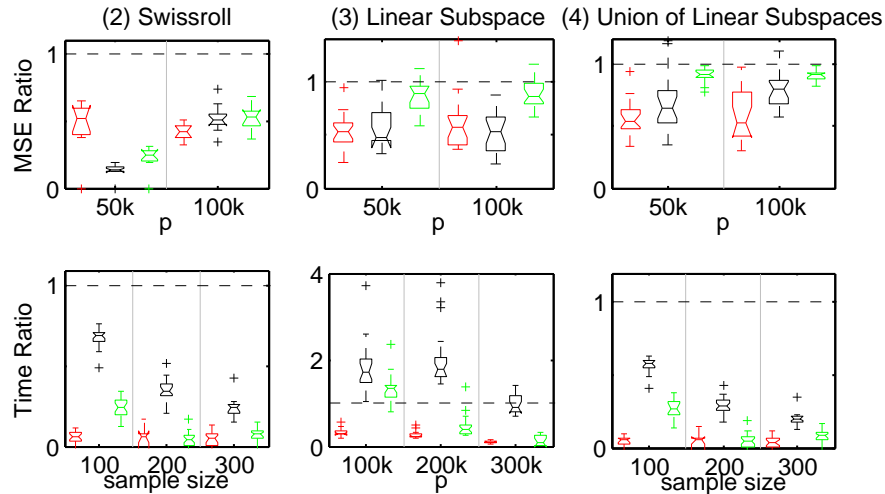

Figure 3: Numerical results for various simulation scenarios. Top plots depict the relative mean-squared error of MSB (our approach), versus CART (red), Lasso (black), and PC regression (green) for as a function of ambient dimension of $x$. Bottom plots depict the ratio of CPU time as a function of sample size. The three simulation scenarios are: swissroll (left), linear subspaces (middle), union of linear subspaces (right). MSB outperforms both CART, Lasso, and PC regression in all three scenarios regardless of ambient dimension ($r_{mse}^{\mathcal{A}} < 1$ for all $p$). MSB compute time is relatively constant as $n$ or $p$ increase, whereas Lasso's compute time increases, thus, as $n$ or $p$ increase, MSB CPU time becomes less than Lasso's. MSB was always significantly faster than CART and PC regression, regardless of $n$ or $p$. For all panels, $n = 100$ when $p$ varies, and $p = 300k$ when $n$ varies, where k indicates 1000, e.g., $300k = 3 \times 10^5$.

Table 1: Neuroscience application quantitative performance comparisons. Squared error predictive accuracy per subject (using leave-one-out) was computed. We report the mean and standard deviation (s.d.) across subjects of squared error, and CPU time (in seconds). We compare multiscale stick-breaking (MSB), CART, Lasso, random forest (RF), and PC regression. MSB outperforms all the competitors in terms of predictive accuracy and scalability. Only MSB and Lasso even ran for the $\approx 10^6$ dimensional application. **Bold** indicates best MSE, $*$ indicates best CPU time.

| DATA | $n$ | $p$ | MODEL | MSE (S.D.) | TIME (S.D.) |
|---|---|---|---|---|---|
| CREATIVITY | 108 | 2,415 | **MSB** | **0.56 (0.85)** | **1.1 (0.02)** |
| | | | CART | 1.10 (1.00) | 0.9 (0.01) |
| | | | LASSO$^*$ | 0.63 (0.95)$^*$ | 0.40 (0.10)$^*$ |
| | | | RF | 0.57 (0.90) | 78.2 (0.59) |
| | | | PC REGRESSION | 0.65 (0.88) | 0.46 (0.37) |
| MOVEMENT | 56 | $\approx 10^6$ | **MSB**$^*$ | **0.76 (0.90)**$^*$ | **20.98 (2.31)**$^*$ |
| | | | LASSO | 1.02 (0.98) | 96.18 (9.66) |

## 5.3 Quantitative Comparisons for Neuroscience Applications

Table 1 shows the mean and standard deviation of point-estimate predictions per subject (using leave-one-out) for the two neuroscience applications that we investigated: (i) predicting creativity from diffusion MRI (creativity) and, (ii) predicting head motion based on functional MRI (movement). For the creativity application, $p$ was relatively small, "merely" $2,415$, so we could run Lasso, CART, and random forests (RF) [30]. For the movement application, $p$ was nearly one million.

For both applications, MSB yielded improved predictive accuracy over all competitors. Although CART and Lasso were faster than MSB on the relatively low-dimensional predictor example (creativity), their computational scaling was poor, such that CART yielded a memory fault on the higher-dimensional case, and Lasso required substantially more time than MSB.

## 6 Discussion

In this work we have introduced a general formalism to estimate conditional distributions via multiscale dictionary learning. An important property of any such strategy is the ability to scale up to ultrahigh-dimensional predictors. We considered simulations and real-data examples where the dimensionality of the predictor space approached one million. To our knowledge, no other approach to learn conditional distributions can run at this scale. Our approach explicitly assumes that the posterior $f_{Y|X}$ can be well approximated by projecting $x$ onto a lower-dimensional space, $f_{Y|X} \approx f_{Y|\eta}$, where $\eta \in \mathcal{M} \subset \mathbb{R}^d$, and $x \in \mathbb{R}^d$. Note that this assumption is much less restrictive than assuming that $x$ is close to a low-dimensional space; rather, we only assume that the part of $f_X$ that "matters" to predict $y$ lives near a low-dimensional subspace. Because a fully Bayesian strategy remains computationally intractable at this scale, we developed an empirical Bayes approach, estimating the partition tree based on the data, but integrating over scales and posteriors.

We demonstrate that even though we obtain posteriors over the conditional distribution $f_{Y|X}$, our approach, dubbed multiscale stick-breaking (MSB), outperforms several standard machine learning algorithms in terms of both predictive accuracy and computational time, as the sample size ($n$) and ambient dimension ($p$) increase. This improvement was demonstrated when the $\mathcal{M}$ was a swissroll, a latent subspace, a union of latent subspaces, and real data (for which the latent space may not even exist).

In future work, we will extend these numerical results to obtain theory on posterior convergence. Indeed, while multiscale methods benefit from a rich theoretical foundation [2], the relative advantages and disadvantages of a fully Bayesian approach, in which one can estimate posteriors over all functionals of $f_{Y|X}$ at all scales, remains relatively unexplored.

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
