[Supplementary Material · nipsMSB_supplement.pdf]

# Supplementary material for: Multiresolution dictionary learning for conditional distributions

## 1 Partition Tree Schematic

Figure 1: (i) Multiscale partition of the data. (ii) Path through the tree for $x_i \in \mathbb{R}^p$. (iii) Conditional density of $y_i$ given $x_i$ defined as a convex combination of densities along the path.

## 2 Predictions

Consider the case we want to predict the response $y^*$ for a future observation based on predictors $x^*$ and previous observations $(x^{(n)}, y^{(n)})$ with $x^{(n)} = (x_1, \ldots, x_n)$ and $y^{(n)} = (y_1, \ldots, y_n)$. Because the partitioning strategy that we adopted lacks an elegant out-of-sample embedding function (unlike other paritioning strategies), we adopt a Voronoi expansion procedure by which the new predictors $x^*$ are allocated to $C_{j,k}$'s having the closest centers with respect to $\rho_W$ (we considered the Euclidean distance). Summaries of the predictive density of $y^*$ will be computed as follows:

(i) allocate predictors $x^*$ to $C_{j,k}$'s having the closest centers with respect to $\rho_W$

(ii) run the Gibbs sampler for $S$ iterations, and at the $s$th iteration:

  a) sample parameters $\{\sigma_{j,k_j}^{(s)}, \mu_{j,k_j}^{(s)}, \pi_{j,k_j}^{(s)}\}_{j \in \mathbb{Z}, k_j \in \mathcal{K}_j}$ from the posterior, i.e. $p(.|x^{(n)}, y^{(n)})$

  b) sample $\hat{y}_s^*$ from

$$\sum_{j \in \mathbb{Z}} \pi_{j,k_j(x^*)}^{(s)} \mathcal{N}\left(\mu_{j,k_j(x^*)}^{(s)}, \sigma_{j,k_j(x^*)}^{(s)}\right)$$

(iii) given the sequence $\{\hat{y}_s^*\}_{s=1}^S$, summaries of the predictive density such as mean, variance and quantiles can be computed.

# 3 Graph partitioning algorithm: METIS

An overview on METIS can be found in [1]. Basically, METIS is an algorithm used to partition graphs operating on a dissimilarity matrix. We construct the graph adding an edge between each pair of data points and assigning weight depending on the distance between the two data points [2]. Consider the case we want to clusters points based on covariates information and let $x_j \in \mathbb{R}^p$ the vector of covariates measured for the $j$th sample. The weighted graph construction follows via computing all pairwise distances using $\rho(x_u, x_v) = \|\tilde{x}_u - \tilde{x}_v\|_2$, where $\tilde{x}$ is the whitened $x$ (i.e., mean subtracted and variance normalized). We let there be an edge between $x_u$ and $x_v$ whenever $\mathrm{e}^{-\rho(x_u, x_v)^2} > t$, where $t$ is some threshold chosen to elicit the desired sparsity level. In all our examples we used the Euclidean distance as metric. Given the weighted graph, the tree is constructed recursively applying METIS. Specifically, starting from the coarse scale, subsets were split into two disjoint subsets using METIS. This process continued until the number of observations in the subsets located at the finest scale dropped below some chosen threshold $\gamma$. We chose $\gamma = 5$ in all our applications.

# 4 Synthetic examples

## 4.1 Competitor Algorithms

As we are unaware of other methods that estimate posteriors with such high-dimensional predictors, we compare point estimates of our approach with other regression algorithms. In particular, we elected to compare against lasso, classification and regression trees (CART), Random Forest (RF) and principal component (PC) regression. The lasso regularization parameter and the number of principal components for PC regression were chosen based on the Akaike information criterion (AIC). For all algorithms, standard Matlab packages were utilized.

## 4.2 Additional results

Tables 1, 2 and 3 show results concerning example 2, 3, and 4 in §4.4. Each Table reports mean squared errors and the mean of amount of time necessary to obtain one point predictions. In particular, Table 1 shows results concerning example 3 (linear subspace) for different number of factors ($d = 5, 10$), Table 2 shows results concerning example 4 (union of linear subspaces) for different number of mixture components ($G = 5, 10$), while Table 3 shows results for example 2 (swissroll). As shown, in almost all simulated scenarios, our model is able to perform as well as or better than the model associated to the lowest mean squared error and scales substantially better than others to high dimensional predictors.

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

Table 1: Linear subspace: Mean and standard deviations of squared errors under multiscale stick-breaking (MSB), CART and Lasso for sample size 50 and 100 for different simulation scenarios.

| $p$ | $n$ | | $d = 5$ | | | $d = 10$ | | |
|---|---|---|---|---|---|---|---|---|
| | | | MSB | CART | LASSO | MSB | CART | LASSO |
| $50k$ | 50 | MSE | 0.18 | 0.31 | 0.25 | 0.22 | 0.58 | 0.22 |
| | | STD | 0.32 | 0.30 | 0.42 | 0.24 | 0.54 | 0.30 |
| | | TIME | 3 | 2 | 1 | 3 | 3 | 1 |
| $50k$ | 100 | MSE | 0.18 | 0.27 | 0.26 | 0.20 | 0.41 | 0.52 |
| | | STD | 0.26 | 0.42 | 0.46 | 0.23 | 0.46 | 0.78 |
| | | TIME | 5 | 5 | 2 | 5 | 5 | 1 |
| $100k$ | 50 | MSE | 0.35 | 0.45 | 0.89 | 0.16 | 0.33 | 0.20 |
| | | STD | 0.53 | 0.77 | 1.04 | 0.21 | 0.46 | 0.31 |
| | | TIME | 3 | 25 | 2 | 3 | 27 | 2 |
| $100k$ | 100 | MSE | 0.43 | 0.88 | 0.52 | 0.17 | 0.50 | 0.31 |
| | | STD | 0.59 | 1.29 | 0.70 | 0.24 | 0.75 | 0.49 |
| | | TIME | 7 | 50 | 5 | 7 | 51 | 5 |
| $500k$ | 50 | MSE | 0.11 | 0.16 | 0.15 | 0.83 | 2.26 | 0.92 |
| | | STD | 0.15 | 0.24 | 0.19 | 1.01 | 2.60 | 3.69 |
| | | TIME | 5 | 90 | 11 | 5 | 121 | 10 |
| $500k$ | 100 | MSE | 0.003 | 0.17 | 0.08 | 0.13 | 1.37 | 1.06 |
| | | STD | 0.16 | 0.23 | 0.13 | 1.12 | 1.81 | 1.50 |
| | | TIME | 10 | 214 | 43 | 8 | 227 | 42 |
| $700k$ | 50 | MSE | 1.70 | 1.48 | 1.47 | 0.66 | 1.65 | 1.07 |
| | | STD | 2.18 | 2.47 | 1.63 | 0.87 | 1.49 | 0.95 |
| | | TIME | 6 | 121 | 12 | 7 | 151 | 13 |
| $700k$ | 100 | MSE | 0.69 | 1.36 | 0.82 | 0.78 | 1.52 | 1.43 |
| | | STD | 0.94 | 1.47 | 1.28 | 1.03 | 1.34 | 2.11 |
| | | TIME | 13 | 321 | 41 | 12 | 325 | 44 |

Table 2: Union of linear subspaces: Mean and standard deviations of squared errors under multiscale stick-breaking (MSB), CART and Lasso for different sample sizes for different simulations sampled from a mixture of factor analyzers

| $p$ | $n$ | SIM | $G = 10$ | | | $G = 5$ | | |
|---|---|---|---|---|---|---|---|---|
| | | | MSB | CART | LASSO | MSB | CART | LASSO |
| 50$k$ | 100 | MSE | 0.23 | 0.42 | 0.36 | 0.17 | 0.43 | 0.22 |
| | | STD | 0.34 | 0.59 | 0.43 | 0.18 | 0.69 | 0.23 |
| | | TIME | 5 | 24 | 3 | 7 | 27 | 3 |
| 50$k$ | 200 | MSE | 0.23 | 0.42 | 0.27 | 0.17 | 0.22 | 0.20 |
| | | STD | 0.33 | 0.56 | 0.23 | 0.19 | 0.38 | 0.25 |
| | | TIME | 10 | 51 | 8 | 12 | 56 | 7 |
| 100$k$ | 100 | MSE | 0.67 | 1.35 | 1.32 | 0.15 | 0.17 | 0.22 |
| | | STD | 1.04 | 2.26 | 1.36 | 0.23 | 0.19 | 0.23 |
| | | TIME | 9 | 47 | 6 | 6 | 44 | 5 |
| 100$k$ | 200 | MSE | 0.64 | 1.37 | 0.85 | 0.15 | 0.26 | 0.15 |
| | | STD | 0.95 | 1.77 | 1.29 | 0.24 | 0.42 | 0.24 |
| | | TIME | 15 | 99 | 15 | 11 | 89 | 15 |
| 300$k$ | 100 | MSE | 0.26 | 0.39 | 0.31 | 0.63 | 1.40 | 1.01 |
| | | STD | 0.39 | 0.51 | 0.52 | 0.80 | 1.24 | 1.46 |
| | | TIME | 9.28 | 125 | 18 | 9 | 145 | 17 |
| 300$k$ | 200 | MSE | 0.25 | 0.47 | 0.26 | 0.63 | 1.17 | 0.92 |
| | | STD | 0.36 | 0.88 | 0.43 | 0.80 | 2.11 | 1.04 |
| | | TIME | 15 | 262 | 40 | 13 | 283 | 43 |
| 300$k$ | 300 | MSE | 0.25 | 0.30 | 0.30 | 0.62 | 1.42 | 0.70 |
| | | STD | 0.36 | 0.41 | 0.48 | 0.89 | 1.85 | 0.94 |
| | | TIME | 15 | 463 | 73 | 16 | 465 | 89 |

Table 3: Swissroll: Mean and standard deviations of squared errors under multiscale stick-breaking (MSB), CART and Lasso for different sample sizes for different simulation scenarios.

| $p$ | $n$ | | MSB | CART | LASSO |
|---|---|---|---|---|---|
| 100$k$ | 50 | MSE | 0.24 | 0.44 | 0.25 |
| | | STD | 0.24 | 0.42 | 0.29 |
| | | TIME | 3 | 22 | 2 |
| 100$k$ | 100 | MSE | 0.24 | 0.43 | 0.17 |
| | | STD | 0.26 | 0.55 | 0.22 |
| | | TIME | 6 | 48 | 7 |
| 200$k$ | 50 | MSE | 0.24 | 0.67 | 0.29 |
| | | STD | 0.23 | 0.50 | 0.29 |
| | | TIME | 4 | 38 | 5 |
| 200$k$ | 100 | MSE | 0.25 | 0.78 | 0.33 |
| | | STD | 0.26 | 0.74 | 0.36 |
| | | TIME | 6 | 96 | 13 |
| 500$k$ | 50 | MSE | 0.17 | 0.47 | 0.23 |
| | | STD | 0.23 | 0.43 | 0.22 |
| | | TIME | 5 | 126 | 10 |
| 500$k$ | 100 | MSE | 0.17 | 0.33 | 0.19 |
| | | STD | 0.21 | 0.46 | 0.23 |
| | | TIME | 11 | 230 | 25 |