[Reviews · NeurIPS 2013]

Submitted by Assigned_Reviewer_6

This paper consider the problem of nonparametric estimation of the conditional distribution when the conditioning variable is high-dimensional. The authors construct a tree-based structure for X which is fixed a priori and build a mixture model on the tree (conditional on the observed regressors for an observation).

Quality: The idea is an interesting one and has the potential for working well with many regressor variables. It's unfortunate the reference for the METIS algorithm is missing and that the method is not clearly described. Seems unclear how pairwise distances are unused to construct the partition. For example, it seems to me that, for a fixed level, pairs x_1,x_2 and x_2,x_3 may meet the criteria but x_1,x_3
don't. It would be helpful to describe how the partitions arise. It would also be useful to mention that the author are defining a mixture model (rather than a linear model which is commonly used with wavelets).

There seems to be no justification for the comment "The implication of this is that each scale within a path is weighted to optimize the bias/variance trade-off across scales."

The paper would benefit some discussion of the properties of the stick-beaking prior.

The prior is infinite-dimensional. Is the prior truncated in the inference?

Clarity: The paper was unclear to me in places. How are some examples:
page 4, what is $\Xi$
page 5, I'm not clear what it means that "we let each $\psi_{j,k}$ simply be a Dirac delta function operating only on the $X$'s". Does it just mean that there is no regression on $X$ in the conditional densities.

Originality: There has been little work on applications of flexible conditional distribution estimation with high-dimensional regressors.

Significance: This has the potential to have a substantial impact.
Summary: A very interesting idea. The paper would benefit from some tightening up.

Submitted by Assigned_Reviewer_7

Summary:
This paper addresses the problem of conditional density estimation with a high dimensional input space (p >> n), an important problems as most (if not all) current models for nonparametric conditional density estimation do not scale to high-dimensions. Moreover, datasets with high dimensional inputs but relatively small sample sizes are becoming increasingly common. The model for the conditional density f(y|x) is defined in three stages. First, a tree structure is defined over the input space. Second, given the tree structure, C_{j,k}, the k^th partition of the X space at scale j, is mapped to a lower dimensional space. Third, a local model for the conditional density at each node j,k is selected which only depends on x through its lower dimensional representation. In practice the lower dimension representation is the indicator function 1(x \in C_{j,k}) and the local model is Gaussian, i.e. f_{j,k}(y|x)=N(m_{j,k}, s_{j,k}).
The inference algorithm is “partially Bayesian”, as an empirical Bayes approach is used for the tree structure (which is estimated via the METIS algorithm). Then, a Gibbs sampling algorithm is used for posterior inference of the weights and parameters (m_{j,k}, s_{j,k}) of each node indexed by j,k of the tree.

Strengths:
The authors develop a novel method for high-dimensional conditional density estimation which lacks any competitors.
Performance is quite impressive when compared with LASSO, CART, PC regression (even more so considering the added flexibility of the model).

Weaknesses:
Poor notation and lack of clear representation of the model for the conditional density. The model is stated very vaguely, but then a more specific version is used. The paper would be more readable if the focus was only on the specific version of the model used in practiced and included a clear mathematical description of the model. In equation (1), your model for the conditional density given the tree structure tau, depends on some j,k. In particular, f(y|x,tau) is a weighted average over the local conditional density of all ancestors of j,k. How is this cut-off j,k selected?
In equation (1), your notation C_{j',k'} \in A_{j,k} does not match your definition of the sets, as C_{j',k'} is a subset of the X space and A_{j,k} is a subset of the integers.
Throughout the text the conditional distribution function is used when conditional density should be, leading to some incorrect statements (examples are pg. line 86 and pg. 6 line 322).
The word approximately is used throughout the text, but it may be more sensible to replace approximately with equals and state that these are model assumptions.
What does f_{y|x} represent? The entire collection of conditional densities f_{y|x}=(f_{y|x}(\cdot|x), x\in X) or a specific f_{y|x}(y|x)? Same comments for F_{Y|X}
pg. 4 line 196, there is a missing reference.
pg. 4 equation (2), should proportional be replaced with equals? And the last pi_{j,k}=1-\sum V_{j',k'} (where the sum is over the ancestors of j,k). Otherwise it involves a normalization term which may affect the full conditional of the V_{j,k}.
pg. 5 line 239, the notation k_j(x_i) is used but not introduced.
pg. 5 line 242, Is the full conditional of V_{j,k} correct? Because the likelihood seems to involve a normalization term based on V_{j,k}.
The paper by Bhattacharya, Page, and Dunson (2013) Density Estimation and Classification via Bayesian nonparametric learning of affine subspaces could be an important reference/competitor to consider.

Quality: The work is supported by some impressive examples but the model lacks a clear mathematical explanation.

Clarity: I would suggest removing the vague representation of the model and just focus on the specific model used in practice. Notation needs some cleaning.

Originality: Similar to other regression techniques, the proposed model achieves flexibility through a tree structure; however, its originality comes from flexibility in modelling the entire conditional density not just the mean function, which is achieved by allowing for local conditional densities at each node of the tree. For inference in high-dimensions, they must take an empirical Bayes approach with regards to the tree structure.

Significance: The results could potentially be very important as the allow for conditional density estimation with high-dimensional inputs and experimental results are impressive.
Summary: Overall, this is a nice paper that address an important issue and has impressive performance over existing, more inflexible models (in terms of both run time and accuracy). However, the exact model used needs to be clearly stated mathematical and notation should be cleaned.

Submitted by Assigned_Reviewer_8

This paper proposes a model for conditional density estimation in extremely high dimensions (p=10^6). First, a tree that hierarchically partitions the sample space is constructed deterministically from the data. The hierarchical partition consists of cells C_{j,k} at different resolutions (j) and locations (k). The "edges" in the tree are given prior weights according to stick-breaking, and this helps determine to which cells the Xs are assigned. The Xs have hard assignments to cells but these assignments are sampled. Given an assignment of Xs to cells, the Xs are mapped to a lower dimensional space, and from this embedding a distribution over Y|X is inferred. In this case Y is modeled as Gaussian, but the method is open to other choices. Experiments are performed on several simulated datasets and on neuroimaging data. The proposed method beats several competitors on the synthetic data in both time and accuracy. On the neuroimaging problems the proposed method is competitive with other methods in terms of MSE, and scales up much better to larger problems.
Summary: A tree-based model for conditional density estimation is proposed for large p (~10^6) problems. The model is not very elegant but it provides good experimental results.
Author Feedback

Author rebuttal: Thank you for your useful comments and suggestions on our manuscript. We will certainly modify the manuscript accordingly. Please, find below our response to your comments.

Reply to Reviewer #1:

METIS is described in detail in [1]. METIS operates on a dissimilarity matrix, constructed via pairwise comparisons. For this work, we followed the advice given in [2]. In short, we used an L2 distance scaled by the variances of each of the data points. The partitions arose via recursively partitioning the graph defined by the nearest neighbors according to the above described dissimilarity. We will clarify in the revision.

We will provide a supplementary figure demonstrating how the weights change across scale according to their relative signal for the inference task.

We are using a stick-breaking prior over an infinitely-deep tree, but only go as deep as the data allows; when we run out of data, we truncate the tree.

$\Xi$ is the embedding space, the space in which the data is embedded after dimensionality reduction. We will clarify

That $\psi_{j,k}$ is a Dirac delta function means that there is no regression on $X$ in the conditional densities, rather, we simply obtain the density for all those elements in the cell $C_{j,k}$.


Reply to Reviewer #2:

Thank you for pointing out some inconsistencies in our notation and distributions. We will resolve them all in the revision.

“pg. 4 line 196. Missing reference.” Karypis and Kumar (1999) [1]
“pg. 4 equation (2).” Yes, proportional should be replaced by equal. To guarantee that weights associated to any path (s_1, …, s_L) sum up to 1, we set stick breaking weights corresponding to subsets located at the last leaves equal to one, i.e. V_{L j}=1 for all j (Ishwaran and James, 2001)[3].
“pg. 5 line 239.” k_j(x_i) is a variable indicating the set at level j where x_i has been allocated.
“pg. 5 line 242.” Since weights (pi_{s_1}, …, pi_{s_L}) sum up to 1 for any path (s_1, …, s_L), there is no normalizing term depending on stick breaking weights. Please, refer to Ishwaran and James (2001) [3] for more details on full conditionals.

Reply to Reviewer #3

“Line 216 claims that the embedding function \psi_{j,k} is a Dirac delta”

See above.

“line 63 claim of NP-hardness is not cited or explained”

[4] is a nice review on variable selection. In brief, the problem is NP-hard because there are $2^p$ possible subsets of variables to select (because we can select any $\binom{p}{k}$ for $k=1,...,p$).

Reference:

[1] G. Karypis and V. Kumar (1999) A fast and high quality multilevel scheme for partitioning irregular graphs. SIAM Journal on Scientific Computing.

[2] W. K. Allard G. Chen and M. Maggioni (2012) Multi-scale geometric methods for data sets II: Geometric multi-resolution analysis. Applied and Computational Harmonic Analysis.

[3] H. Ishwaran and L. F. James (2001), Gibbs Sampling Methods for Stick-Breaking Priors, JASA.

[4] I. Guyon, A. Elisseeff (2003) An Introduction to Variable and Feature Selection, JMLR.